# Firm Performance during COVID-19 Pandemic: Does Ownership Identity Matter? Evidence from Indonesia

Dian Perwitasari *, Doddy Setiawan , An Nurrahmawati and Isna Putri Rahmawati

Accounting Study Program, Faculty of Economics and Business, Sebelas Maret University, Surakarta 57126, Indonesia
* Correspondence: dianperwitasari@staff.uns.ac.id

**Abstract:** This study aimed to examine the importance of shareholder identity in improving company performance during shock events such as the COVID-19 pandemic. The outbreak poses threats and opportunities for businesses in various countries including Indonesia. Subsequently, companies must adapt to address the consequences of the economic disruption and lockdown policies imposed by the local government. The study sample comprised companies listed on the Indonesia Stock Exchange (IDX) during the COVID-19 pandemic from 2020 to 2021. Fixed effects model regression was employed to examine the effect of family, government, and institutional ownership on company performance. The results showed that family and institutional ownership positively affected company performance during the pandemic. The mechanisms of direct supervision and control by family members could potentially increase the benefits of their businesses. Furthermore, high institutional ownership makes the role of investors substantial in reducing business risk and increasing company performance. Furthermore, the results revealed that government ownership negatively affected company performance. As owners, the government has different strategic objectives, where companies are more oriented toward better public services than financial gains. Therefore, it is essential to consider the impact of shareholder involvement on company performance, especially during a pandemic because they are treated differently. The research suggests that organizations are responding and adapting to the uncertainties in the business environment they face through a variety of mechanisms, including developing public and corporate governance strategies to prepare for and respond to future emergencies.

**Keywords:** company performance; ownership structure; during COVID-19 pandemic; panel data model; Indonesia

## 1. Introduction

Previous studies have focused on the consequences of catastrophic events on firm performance, for example, the studies that have focused on the impact of hurricane events (Schuwer et al. 2019), (Dessaint and Matray 2017); bushfires (Siddikee and Rahman 2017); and financial crisis (Huang et al. 2020), (D'Aurizio et al. 2015), (Lins et al. 2017), (Clarke et al. 2012), (Mitton 2002), (Rajan and Zingales 1998). On 12 March 2020, the World Health Organization (WHO) confirmed the coronavirus disease (COVID-19) outbreak as a pandemic owing to its global impact on various sectors, including health and the real economy. According to the Asian Development Bank (2020), the potential impact of the pandemic on the global economy is between USD 5.8 trillion and 8.8 trillion. This is equivalent to 6.4–9.7% of the world's gross domestic product (Zheng and Zhang 2021). The social restriction policy to reduce transmission and prevent the spread of the virus has caused a sharp and immediate decline in economic production in various countries. The resulting economic crisis differs from previous crises in terms of binding financial performance challenges, value chain disruptions, and difficulties conducting business activities during restrictions (Amore et al. 2022; Ding et al. 2021).

Many studies have attempted to determine the key variables that affect company performance during the COVID-19 pandemic (Amore et al. 2022), (El-Chaarani et al. 2022), (Engidaw 2022), (Shan and Tang 2022), (Ding et al. 2021), (Golubeva 2021), (Kaczmarek et al. 2021); (Khatib and Nour 2021), (Li et al. 2021), (Wu and Xu 2021), (Albuquerque et al. 2020), (Bartik et al. 2020), (Fahlenbrach et al. 2020), (Jabbouri and Jabbouri 2020), (Kraus et al. 2020), (Levy 2020), (Obrenovic et al. 2020), (Shen et al. 2020). Obrenovic et al. (2020) investigated the key factors influencing company operations and the initial strategies for dealing with adversity during a pandemic. The study finds that substantial heterogeneity depends on the company, industry, and macroeconomic factors of the country. Firms with higher cash reserves and lower debt are more robust to the effects of COVID-19 (Ding et al. 2021), (Fahlenbrach et al. 2020). The pandemic has provided opportunities for companies in certain industries to innovate and bring new products (Li et al. 2021). This presents an interesting scenario in which a surprise drastically changes company performance, shifting managers' expectations in the months before the crisis (Golubeva 2021), (Obrenovic et al. 2020).

Since the spread of COVID-19, companies have closely monitored the market trends of their operations. PT Hero Supermarket Tbk (HERO), Indonesia's giant retail company, was affected by a substantial change in customer spending patterns due to the lockdown. HERO recorded insignificant performance throughout the first semester of 2021. The company lost IDR550.88 billion compared to the previous year's IDR202.07 billion (Liputan6.com 2021). Companies with good governance respond quickly to new macroeconomic fluctuations (Huang et al. 2020). However, other studies have shown conflicting views on the relationship between company performance and corporate governance mechanisms (Ahsan et al. 2021), (Chijoke-Mgbame et al. 2020). The shareholder-centered governance model suggests that separating a company's ownership from its control causes agent–principals conflict. This is because managers prioritize their profits by maximizing shareholder value (Jensen and Meckling 1976), (Dey 2008).

Over the last few decades, the principal–agent issue has been fundamental in modern finance. Agents are trusted to act in the principal's interest, but they prioritize their interests (Jabbouri and Jabbouri 2020). A lack of oversight allows agents to expropriate corporate resources by stealing profits and channeling company assets through transactions with related parties or other companies (Chan 2010), (La Porta et al. 2000). Furthermore, agents recruit unqualified relatives in key positions and determine excessive salaries (Schulze et al. 2001). Xu and Wang (1999) examined ownership government, institutional, and individual influences on company performance in China. The study finds that large institutional shareholders increase company performance. However, increased government ownership decreases labor productivity, resulting in inefficient company performance. The different shareholders' characteristics, behavior, focus, and strategies describe their involvement in the company and its impact on its performance.

According to a press release by the Ministry of Industry on 21 September 2021, Indonesia's economic strength lies in its large domestic market. It has superior manufacturing value added to other ASEAN countries such as Thailand (USD 1.23 billion), Malaysia (USD81.19 million), and Vietnam (USD41.7 Million). Therefore, Indonesia is the largest economy in Southeast Asia (Kemenperin.go.id 2021). The Report Indonesia 2020 reports real gross domestic product growth in 2019 of 5.03% and before the COVID-19 outbreak, the gross domestic product was expected to increase in 2020 (Oxford Business Group 2020). However, the COVID-19 pandemic forced Bank Indonesia to revise 4.2–4.6% of its 2020 economic growth projection under the 2019 growth rate. According to the East Ventures Digital Competitiveness Index 2021 Report, Indonesia achieved the fastest digital economic growth in the ASEAN region during the pandemic because its regulatory framework is more open than that of other markets in Southeast Asia (East Ventures Digital 2021).

The threats and opportunities arising in Indonesia are interesting phenomena because they relate to how ownership identity affects company performance. They also facilitated economic disruption due to the spread of COVID-19 and lockdowns worldwide. Indonesia as one of the largest emerging markets in Southeast Asia has potential for new and re-

newable energy (EBT) and export-oriented manufacturing. This distinguishes the country from neighboring countries, increasing its attractiveness to foreign investors. Moreover, the Indonesia Stock Exchange (IDX) was the most active in the ASEAN region during Q3-2021 (EY Global IPO Trends Reports 2021). As a member of the Corporate Governance Task Force, the IDX prepares a corporate governance roadmap to increase investment transparency and attract and protect foreign investors' interests. Although this step is fundamental to addressing agency problems, its implementation remains challenging for securities market regulators. Setiawan et al. (2016) examine Indonesia's corporate ownership identity and dividend policy. The study finds that family firms negatively affect dividend payments by making lower payments because they want to be in control, benefiting from these resources at the expense of minority shareholders. Therefore, the family company is involved in a takeover by a minority shareholder because of poor investor protection, lack of transparency, as well as information disclosure. It indicates that Indonesia has the seriousness of the IDX agency problem.

In addition, we believe the issue regarding various ownership roles in the firm performance needs a deeper analysis, especially in emerging markets. Most empirical studies on the impact of ownership identity on company performance have focused on developed countries such as the United States (Admati et al. 1994), (Hollowell 2006), (Villalonga and Amit 2006), the United Kingdom (Becht et al. 2008), and Japan (Yoshikawa and Rasheed 2010). They highlighted the role of governance mechanisms in improving company performance. Only a few studies have examined the relationship between ownership structure and company performance in emerging markets such as Indonesia, especially during the COVID-19 pandemic. Therefore, they cannot guide companies in emerging markets in making business decisions because the uniqueness of emerging markets itself. As such, countries have limited disclosure, inadequate investor protection, and loose regulations (Yu 2013). This study will expand its scope in terms of; (1) the research period used is the COVID-19 pandemic since the reduction in economic activities that happened during this period may lead to a decline in the firm performance; (2) include control variables such as leverage, business risk, liquidity, firm size, growth opportunities, and sector; (3) and investigation of companies in one of the emerging markets in the ASEAN region with the most active capital market in the ASEAN region. The data will be analyzed using panel data regression with an analytical approach to support this novelty.

This study aims to fill a research gap by addressing the following research question: Does owner identity play a significant role in business performance during the COVID-19 pandemic? This research tries to answer the question in Indonesia, a developing country with different characteristics from developed countries.

The following sections first provide an overview of the theoretical background, literature survey, and hypothesis development. Data and methodology are then described in the third section. The fourth section describes experimental results and robustness tests. Finally, the fifth section presents conclusions, limitations, and implications.

## 2. Literature Review and Hypothesis Development

### 2.1. Theoretical Background

The 1972 theory by Alchian and Demsetz of property rights treats a firm as a contractual entity in which managers select workers and determine their duties. They state that insiders should have ownership and control rights. Similarly, the rights to production factors should be distributed to different stakeholders, provided that the income from the provision of the rights exceeds costs. Jensen and Meckling (1976) proposed the agency theory by referring to the contractual relationship between the principal and agent. This theory explains inherent conflicts of interest between shareholders (principals) and managers (agents). This conflict occurs because of differences in interests between the principal and the agent. Blockholding has been noted in the corporate governance literature as an effective approach for minimizing agency problems between shareholders and managers (Jensen and Meckling 1976), (Shleifer and Vishny 1997), (Claessens and Djankov 1999).

Block shareholders offer a partial solution to the small-investor free-rider problems. However, blockholder ownership exceeding a certain level causes the owner–manager camp to expropriate minority shareholder wealth (Fama and Jensen 1983), (Morck et al. 1989), (Claessens et al. 1999), so incurring substantial costs associated with blockholders. For instance, this is called the entrenchment problem.

One of the aspects of agency theory is entrenchment theory. It arises from the possibility that large shareholders choose to use their power to take-over minority shareholders by taking investment actions and decisions to fulfill their interests (La Porta et al. 2000), (Claessens et al. 1999). In addition to that, an entrenchment problem might happen when managers utilize information asymmetry to make investment decisions to ensure that their position is irreplaceable (El-Chaarani et al. 2022). Knowledge of a company's financial condition allows managers to know about underutilized cash sources, such as funding for dormant projects or canceled insurance policies. These cash sources form an expense account or are used to fund golden parachutes. Another entrenchment practice is the poison pill, involving the prevention of ownership dilution by offering more shares to shareholders. In addition, managers implement supermajority amendments, requiring a vote by management to approve key measures (El-Chaarani et al. 2022). Performance-based compensation, board independence, and ownership concentration are internal mechanisms to control entrenchment, while external mechanisms include legal protection (El-Chaarani et al. 2022), (Jensen and Meckling 1976).

### 2.2. Literature Review and Hypothesis Development

COVID-19 has rocked businesses worldwide, testing the resilience of various companies to the pandemic. Indonesia had the highest number of infections in Southeast Asia. On 11 July 2022, the country recorded 1007 deaths, which is the highest daily death record in the world. This was followed by Russia, India, and Brazil, with 749, 720, and 597 cases, respectively (Cnnindonesia.com 2021). The pandemic has also significantly impacted the economic, political, social, cultural, defense, and security fields, as well as the welfare of the Indonesian people (Alam et al. 2021).

The COVID-19 epidemic has had an impact on many facets of life across the world, including the economy. According to Pourmansouri et al. (2022), factors of Good Corporate Governance determined how the company's value performance altered during Covid-19. The study's statistical sample includes 120 firms that were listed on the Tehran Stock Exchange between 2011 and 2021. The findings revealed that concentration of ownership is averse to the adoption of corporate governance (GCG) standards. Centralized voter ownership, in particular, weakens the corporate governance structure (CGS). According to the findings of this panel investigation, the concentration of ownership has an impact on the quality of his CGS, making it hard for major shareholders to oppose the power of large shareholders. It also had a detrimental influence on the quality of corporate boards.

Ownership structure affects company performance, providing important information for users of financial statements when making economic decisions. Company performance is often measured using accounting- and market-based approaches (Al-Matari et al. 2014). Return on Assets (ROA) is commonly used in accounting-based measurements (Al-Matari et al. 2014). This metric reflects a company's efficiency in using assets in operational activities to meet shareholders' economic interests. This ratio assesses the efficiency of an entity's assets in generating profits. Assets are business assets acquired with owner capital through the sale of shares on the stock market and/or through debt owed to lenders or other third parties (including the issuance of bonds). Profit assesses the effectiveness of management in terms of revenue and operational cost-effectiveness. Assessing an asset's ability to produce a profit is significant because it has a favorable impact on a company's ability to generate positive cash flow and give investment returns to its shareholders. Managers' and shareholders' interests can be aligned by ownership type (Laporšek et al. 2021). (Laporšek et al. 2021) show that perceptions of corporate objectives vary widely depending on the ownership type in Slovenian companies. The right combination of ownership types

increases the chance of good corporate governance. Consequently, they increase efficiency and firm value, facilitate access to capital markets and funds, and reduce capital costs (Claessens and Yurtoglu 2013).

Alchian and Demsetz (1972) and Jensen and Meckling (1976) define a firm as a contract between the factors of production that act in their self-interest. The success of production factors depends on team performance, the environment in which they operate, and competition with other organizations (Fama 1980). Therefore, a company's performance results from its operating activities, including its impact on all interest groups. Stakeholders, comprising owners, management, employees, suppliers, customers, local communities, states, and others, aim to improve a company's performance. In this case, performance is the profitability and rate of return on the stakeholders' investment, depending on their involvement in the company and their ability to diversify risk.

### 2.2.1. Family Ownership

The literature explains that family involvement in business improves companies' performance (Jabbouri and Jabbouri 2020), (Ding et al. 2021), (Adams et al. 2008). Therefore, a study on how family ownership responds to the COVID-19 shock helps understand the pandemic's general consequences on the business sector (Amore et al. 2022). The implications of family control on the ability to cope with a pandemic are unclear (Amore et al. 2022). The desire to inherit healthy businesses is the most frequent motivation for family owners. (Adams et al. 2008). Subsequently, family companies exhibit longer time horizons in decision-making, greater involvement and reputation problems, and more business strength than non-family owners. During a crisis, families engage in actions aimed at defending their interests at the expense of minority investors, thereby reducing firm value (Claessens and Yurtoglu 2013), (Claessens et al. 1999). The owner's low propensity to make workforce adjustments may be a disadvantage because managers cannot make the complex organizational changes needed to cope with a shock event (Pérez-González 2006).

Family owners' motivations are more valuable during a pandemic because they signal to investors and stakeholders that the business would strive to thrive. Previous studies have also shown that family companies are better prepared to deal with growing political instability (Amore and Minichilli 2018). They have a network of connections that assists them in accessing funding (D'Aurizio et al. 2015). Furthermore, such companies exhibit higher employee productivity (Amore et al. 2022), (Sraer and Thesmar 2007) and lower debt costs (Anderson and Reeb 2003). This is because of stronger relationships with stakeholders and better institutional alignments. Therefore, family companies should be better prepared to disrupt COVID-19.

**Hypothesis 1.** *Family ownership significantly and positively affects company performance.*

### 2.2.2. Government Ownership

Higher government ownership improves company performance because government companies have clear mechanisms and rules. Companies are less constrained by government policies than non-government companies. Furthermore, companies with government ownership easily access monopolies and are more likely to receive government subsidies or low-interest loans when in trouble (Wu and Xu 2021; Berkowitz et al. 2017). Government ownership improves company performance at a certain threshold. First, a government-dominated shareholding structure may be more effective than dispersed shareholding in reducing agency costs where law enforcement is weak. This is illustrated by Monkey King Limited's case, where a large shareholder's absence to monitor its activities enabled the company to manipulate accounts and squander its cash flow. However, this can be avoided when there is concentrated share ownership from any source including the government (Tian and Estrin 2008).

Company performance can also decrease through government share ownership (Yu 2013), (Shleifer and Vishny 1997). As the government is a majority shareholder, it

may pursue goals unrelated to profit maximization (Tian and Estrin 2008). Several studies have shown that government ownership is detrimental to a company's performance. According to Shleifer and Vishny (1997), investors prefer non-governmental ownership to governmental ownership. This is due to the government having a grabbing hand that extorts companies to benefit both politicians and bureaucrats, such as deliberately diverting company resources to political supporters. In addition, most government-owned shares are non-tradable, which is a major obstacle to domestic market development. Wu and Xu (2021) compared the performance of SOE and NSOE in China during the COVID-19 pandemic. The results show that SOE companies receive greater government support. However, they bear more of the burden imposed by the government and are driven less by market signals. This is because most companies have concentrated ownership structures, limited information disclosure, weak investor protection, and dependence on the banking system (Yu 2013).

During the COVID-19 pandemic, the government hopes that the companies they control can survive. Government ownership helps companies survive financial crises by injecting capital (Li et al. 2008). In line with this, Sun et al. (2003) emphasize that the government has a strong financial interest in the successful performance of companies in which it is the majority shareholder. During the pandemic, the government helped companies through preferential treatment. Therefore, state-owned companies may be better prepared for the COVID-19 disruption.

**Hypothesis 2.** *Government ownership significantly and positively affects company performance.*

### 2.2.3. Institutional Ownership

Several studies have shown a positive relationship between institutional ownership and company performance (Jabbouri and Jabbouri 2020), (Amin and Hamdan 2018), (Abu-Serdaneh et al. 2010). Proactive involvement entails writing open letters to management or directors, inviting major shareholders to public meetings, requesting special disclosures or audits, and voting on the feasibility of business decisions (Larcker and Tayan 2011). Companies with significant institutional ownership attract more analysts because institutional oversight allows them to experience lower information asymmetry and better financial performance (Boone and White 2015), (Demiralp et al. 2011). Owing to the high supervision cost, only large shareholders, such as institutional investors, have sufficient interest and incentives to supervise, discipline, and influence managers. This causes managers to focus on company performance without behaving opportunistically. However, some studies show that institutional investors emphasize their gains at the expense of minority shareholders (La Porta et al. 2000), (Claessens et al. 1999). These studies fail to confirm the active involvement of institutional ownership in maximizing shareholder value. This was due to the absence of active institutional supervision of management performance, which required special skills and expertise.

Jabbouri and Jabbouri (2020) show that investors are more likely to recognize the importance of the supervisory role of institutions in Moroccan companies during an economic crisis. According to Rajan and Zingales (1998), regulators and investors ignore corporate governance mechanisms during good economic times in East Asian markets. Shareholders are deeply concerned about corporate governance during economic downturns when performance declines and corporate resources become more accessible to insider's takeover (Mitton 2002). Therefore, institutionally owned companies may be better prepared for COVID-19 disruptions.

**Hypothesis 3.** *Institutional ownership significantly and positively affects company performance.*

### 3. Methodology

The study sample comprised all non-financial companies listed on the Indonesia Stock Exchange for the 2020–2021 period during the COVID-19 pandemic. The financial sector

companies are excluded since it has a distinct business nature. Because the nature of banks and insurance companies is regulated, their underlying characteristics differ from non-financial companies. Non-financial companies bring together economic units that share many characteristics and behavior. The final sample consisted of 1025 firm-year observations. The financial report data of the company were obtained from the IDX and the company's websites. The dependent variable was the company's performance (PERFORM), while the independent variables included family ownership (FAMOWN), government ownership (GOVOWN), and institutional ownership (INSTOWN). The control variables used were growth opportunities (GROWTH), leverage (LEV), liquidity (LIQUID), business risk (BUSRISK), firm size (SIZE), and the industrial sector (SECTOR). Table 1 presents the definition of the dependent, independent, and control variables.

**Table 1.** Definition of variables.

| No | Variable | Abbreviation | Formula | Source |
|----|----------|--------------|---------|--------|
| 1 | Firm Performance | PERFORM | Return on assets ratio | (Amin and Hamdan 2018) |
| 2 | Family Ownership | FAMOWN | Percentage of number of shares held by the family divided by the total number of shares outstanding | (Amore et al. 2022), (Jabbouri and Jabbouri 2020), (Villalonga and Amit 2006) |
| 3 | Government Ownership | GOVOWN | Percentage of number of shares held by the government divided by the total number of shares outstanding | (Yu 2013) |
| 4 | Institutional Ownership | INSTOWN | Percentage of number of shares held by the institutions divided by the total number of shares outstanding | (Jabbouri and Jabbouri 2020), (Boone and White 2015) |
| 5 | Leverage | LEV | Debt to total assets ratio | (Amore et al. 2022) |
| 6 | Business Risk | BUSRISK | EBIT standard deviation | (Wiagustini and Pertamawati 2015) |
| 7 | Liquidity | LIQUID | Current ratio | (Jabbouri and Jabbouri 2020) |
| 8 | Firm Size | SIZE | Natural Logarithm of Total Assets | (Setiawan et al. 2016) |
| 9 | Growth Opportunities | GROWTH | Asset growth ratio | (Jabbouri and Jabbouri 2020) |
| 10 | Industrial Sector | SECTOR | Dummy for industrial sectors | (Samasta et al. 2018) |

Source: Author's computational results.

This study examines the effect of ownership identity on company performance during the COVID-19 pandemic using the following equation:

$$\text{PERFORM}_{it} = \alpha_i + \beta_1 \text{FAMOWN}_{it} + \beta_2 \text{GOVOWN}_{it} + \beta_3 \text{INSTOWN}_{it} + \beta_4 \text{GROWTH}_{it}\, \beta_5 \text{LEV}_{it} + \beta_6 \text{BUSRISK}_{it} + \beta_7 \text{LIQUID}_{it} + \beta_8 \text{SIZE}_{it} + \beta_9 \text{SECTORdummies}_{it}$$

This study uses a panel data regression analysis using the fixed effect model (FEM), the common effect model (CEM), and the random effect model (REM). First, determine the best model to use among these models. The Chow test is applied to determine which model is the best between the common effect model (CEM) and the fixed effect model (FEM). In addition, the Hausman test was conducted to determine which model is best used between the fixed effect model (FEM) and the random effect model (REM).

## 4. Results

### 4.1. Descriptive Statistics

Table 2 demonstrates that the average performance is 0.0625, implying that most companies had positive profitability during the 2020–2021 period. This indicates that

they generated profits, showing their resilience even during the pandemic. The positive value can be attributed to the company's rapid response to the COVID-19 crisis. Many companies changed their business activities to enhance efficiencies with rapid technology adoption and alternative income sources (Obrenovic et al. 2020). Furthermore, the average percentage of family ownership is 50.59%, signifying that more than half of the companies are family owned, with a high median value of 58.08%. Claessens et al. (2000) states that companies in Indonesia have a concentrated ownership structure with dominant family control. Government ownership has an average percentage of 1.41%, with 0% and 82.48% as the maximum and maximum values, respectively, meaning that most companies are not state-owned. This is consistent with the median value of 0% for government ownership. Institutional ownership averages 61.39%, meaning that the sample companies have high institutional ownership. According to Amin and Hamdan (2018), the ability of institutions to influence company decisions depends on their share ownership. Table 2 also lists the control variables.

**Table 2.** Descriptive statistics.

|  | PERFORM | FAMOWN | GOVOWN | INSTOWN | GROWTH | LEV | BUSRISK | LIQUID | SIZE | SECTOR |
|---|---|---|---|---|---|---|---|---|---|---|
| Mean | 0.0625 | 50.5909 | 1.41373 | 61.3919 | 0.70353 | 3.02677 | 28.9580 | 6.3498 | 28,367.12 | 5.7795 |
| Median | 0.0500 | 58.0800 | 0.00000 | 66.8600 | 0.02000 | 0.45000 | 29.0000 | 1.5880 | 28,341.00 | 6.0000 |
| Maximum | 0.5320 | 99.0040 | 82.4800 | 99.9400 | 473.706 | 519.369 | 30.0000 | 1026.0 | 33,537.00 | 9.0000 |
| Minimum | −0.5190 | 0.00000 | 0.00000 | 0.00000 | −0.99900 | 0.00100 | 23.0000 | −0.6850 | 22,531.00 | 1.0000 |
| Std. Dev. | 0.1271 | 30.8038 | 9.32958 | 26.0278 | 15.0101 | 32.8664 | 0.46054 | 42.802 | 1842.473 | 2.5610 |
| Observation | 1025 | 1025 | 1025 | 1025 | 1025 | 1025 | 1025 | 1025 | 1025 | 1025 |

Source: Author's computational results.

### 4.2. Correlation

The correlation matrix between the variables in Table 3 shows a weak pairwise correlation among all independent variables. Table 3 also shows that the correlation coefficient between all independent variables on the dependent variables, on average, is weak. This can be seen from the value of each correlation, FAMOWN 0.0173, GOVOWN −0.0657, INSTOWN 0.0945, GROWTH −0.0141, LEV −0.0084, BUSRISK 0.0224, LIQUID −0.0334, SIZE 0.2356, SECTOR −0.1265. This implies that the sample is free of multicollinearity.

**Table 3.** Correlation matrix of the variables.

| Corelation Probability | 1 | 2 | 3 | 4 | 5 | 6 | 7 | 8 | 9 | 10 |
|---|---|---|---|---|---|---|---|---|---|---|
| PERFORM | 1.0000 | | | | | | | | | |
| FAMOWN | 0.0173 | 1.000 | | | | | | | | |
| GOVOWN | −0.0657 | −0.0307 | 1.0000 | | | | | | | |
| INSTOWN | 0.0945 | 0.0477 | −0.0502 | 1.0000 | | | | | | |
| GROWTH | −0.0141 | 0.0229 | 0.0014 | 0.0153 | 1.0000 | | | | | |
| LEV | −0.0084 | 0.0298 | −0.0018 | 0.0222 | −0.0032 | 1.0000 | | | | |
| BUSRISK | 0.0224 | −0.0176 | 0.0031 | −0.0213 | 0.0020 | 0.0073 | 1.0000 | | | |
| LIQUID | −0.0334 | 0.0384 | −0.0049 | −0.1232 | −0.0049 | −0.0105 | −0.0242 | 1.0000 | | |
| SIZE | 0.2356 | −0.2227 | 0.0784 | −0.0091 | 0.0114 | −0.0968 | 0.0575 | −0.1122 | 1.0000 | |
| SECTOR | −0.1265 | −0.0290 | 0.0137 | 0.0175 | 0.0195 | 0.0287 | −0.1146 | 0.0182 | −0.1846 | 1.0000 |

Source: Author's computational results.

### 4.3. Regression

The results of panel data testing are shown in Table 4. Model specification testing is carried out first to decide which model is appropriate. Based on Hausman's test, the fixed-effects model seems the most appropriate. Based on the results of the fixed effects model in

Table 4, it can be seen that all independent variables, such as family ownership, government ownership, institutional ownership, and control variables such as growth opportunities, leverage, liquidity, business risk, firm size, and the industrial sector simultaneously has a significant effect on firm performance, F-test = 2.1921; $p$ = 0.00000). Furthermore, the adjusted R square value shows a value of 0.3895 which means the model has a reasonably good predictive ability.

**Table 4.** Data panel results.

| | Dependent Variable: PERFORM | | |
|---|---|---|---|
| | **Fixed Effect Model (FEM)** | **Common Effect Model (CEM)** | **Random Effect Model (REM)** |
| Constant | 0.0864 | −0.4519 | −0.3233 |
| | 0.2836 | (1.803) | (1.367) |
| FAMOWN | 0.0004 * | 0.000258 * | 0.0003 * |
| | 1.5863 | 2.0153 | 1.8562 |
| GOVOWN | $-5.73 \times 10^{-7}$ ** | $-5.18 \times 10^{-7}$ *** | $-5.52 \times 10^{-7}$ *** |
| | (2.4402) | (2.5541) | (2.8938) |
| INSTOWN | 0.0008 *** | 0.0004 *** | 0.0005 *** |
| | 2.4534 | 3.0222 | 3.0902 |
| GROWTH | 0.0006 ** | −0.0002 | $6.56 \times 10^{-5}$ |
| | 1.9675 | (0.5995) | 0.274 |
| LEV | 0.0003 * | $5.14 \times 10^{-5}$ | $9.49 \times 10^{-5}$ |
| | 1.5781 | 0.4401 | 0.8003 |
| BUSRISK | −0.105 | 0.0007 | −0.0030 |
| | (1.1028) | 0.0854 | (0.3796) |
| LIQUID | $8.83 \times 10^{-05}$ | $1.19 \times 10^{-5}$ | $2.53 \times 10^{-5}$ |
| | 0.6728 | 0.1309 | 0.2784 |
| SIZE | $1.00 \times 10^{-5}$ ** | $1.68 \times 10^{-5}$ *** | $1.61 \times 10^{-5}$ *** |
| | 2.3874 | 7.6383 | 7.0131 |
| SECTOR | −0.0129 *** | −0.0040 *** | −0.0050 *** |
| | (3.9221) | (2.5950) | (3.0285) |
| Adjusted $R^2$ | 0.3895 | 0.0743 | 0.0685 |
| Prob. Chow Test | 0.0000 | | |
| Prob. Hausman Test | | | 0.0157 |
| N | 1025 | 1025 | 1025 |

Note: ***, **, * indicate a significance level of 1%, 5%, 10%, respectively. The numbers mentioned representing the coefficient value of the variable. On the other hand, the values in brackets represent the values of the t-statistic. Fixed effect model was selected based on the Chow test and Hausman test. Source: Author's computational results.

We also further confirm our statistical results from Table 4, we divided our research model into 4 equation models. By using the fixed effect model (FEM), Table 5 shows the results of testing the effect of 1 independent variable and all control variables on the dependent variable, which we call models 1, 2, and 3. While model 4 is a regression test to test the effect of all independent variables and the control variable on the dependent variable. Our results are consistent. We checked using the following regression equation model:

**Table 5.** Regression Results.

| Variable | Model 1 | | | Model 2 | | | Model 3 | | | Model 4 | | |
|---|---|---|---|---|---|---|---|---|---|---|---|---|
| | Coeff. | Std. Error | T-Value | Coeff. | Std. Error | t-Value | Coeff. | Std. Error | t-Value | Coeff. | Std. Error | t-Value |
| C | 0.1424 | 0.3065 | 0.4646 | 0.1377 | 0.3062 | 0.4497 | 0.0824 | 0.3061 | 0.2693 | 0.0863 | 0.3045 | 0.2836 |
| FAMOWN | 0.0005 | 0.0002 | 1.8980 ** | | | | | | | 0.0004 | 0.0002 | 1.5862 * |
| GOVOWN | | | | $-6.12 \times 10^{-7}$ | $2.36 \times 10^{-7}$ | $-2.5902$ *** | | | | $-5.73 \times 10^{-7}$ | $2.35 \times 10^{-7}$ | $-2.4402$ ** |
| INSTOWN | | | | | | | 0.0008 | 0.0003 | 2.7494 *** | 0.0007 | 0.0003 | 2.4534 *** |
| GROWTH | 0.0001 | 0.0001 | 0.7743 | 0.0005 | 0.0003 | 1.9315 ** | 0.0001 | 0.0001 | 0.7850 | 0.0005 | 0.0002 | 1.9674 ** |
| LEV | 0.0002 | 0.0001 | 1.5105 * | 0.0002 | 0.0001 | 1.4458 * | 0.0002 | 0.0001 | 1.4835 * | 0.0002 | 0.0001 | 1.5780 * |
| RISKBUS | $-0.0108$ | 0.0096 | $-1.1221$ | -0.0098 | 0.0095 | $-1.0315$ | $-0.0097$ | 0.0095 | $-1.0164$ | $-0.0104$ | 0.0095 | $-1.1028$ |
| LIQUID | $3.19 \times 10^{-5}$ | 0.0001 | 0.2435 | $4.21 \times 10^{-5}$ | 0.0001 | 0.3234 | 0.0001 | 0.0001 | 0.7685 | $8.83 \times 10^{-5}$ | 0.0001 | 0.6728 |
| SIZE | $9.83 \times 10^{-6}$ | $4.13 \times 10^{-6}$ | 2.3817 ** | $9.97 \times 10^{-6}$ | $4.22 \times 10^{-6}$ | 2.3619 ** | $9.93 \times 10^{-6}$ | $4.11 \times 10^{-6}$ | 2.4157 ** | $1.00 \times 10^{-5}$ | $4.19 \times 10^{-6}$ | 2.3873 ** |
| SECTOR | $-0.0125$ | 0.0033 | $-3.7705$ *** | $-0.0125$ | 0.0033 | $-3.8049$ *** | $-0.0129$ | 0.0033 | $-3.8974$ *** | $-0.0128$ | 0.0032 | $-3.9220$ *** |
| Adj $R^2$ | 0.3710 | | | 0.3799 | | | 0.3761 | | | 0.3895 | | |
| F-statistic | 2.1054 | | | 2.1489 | | | 2.1298 | | | 2.1921 | | |
| Prob F statistic | 0.0000 | | | 0.0000 | | | 0.0000 | | | 0.0000 | | |
| Observation | 1025 | | | 1025 | | | 1025 | | | 1025 | | |

Note: ***, **, and * significant at 1%, 5%, and 10%; PERFORM = return on assets ratio, FAMOWN = percentage of number of share held by the family divided by the total number of shares outstanding; GOVOWN = percentage of number of share held by the government divided by the total number of shares outstanding; INSTOWN = percentage of number of share held by the institutions divided by the total number of shares outstanding; LEV = debt to total assets ratio; BURISK = EBIT standard deviation; LIQUID = current ratio; SIZE = Ln total assets; GROWTH = asset growth ratio; SECTOR = dummy for nine industrial sectors in Indonesia. Source: Author's computational results.

Model 1

$$PERFORM_{it} = \alpha_i + \beta_1 FAMOWN_{it}\ \beta_2 GROWTH_{it}\ \beta_3 LEV_{it} + \beta_4 BUSRISK_{it} + \beta_5 LIQUID_{it} + \beta_6 SIZE_{it} + \beta_7 SECTORdummies_{it}$$

Model 2

$$PERFORM_{it} = \alpha_i + \beta_1 GOVOWN_{it} + \beta_2 GROWTH_{it} + \beta_3 LEV_{it} + \beta_4 BUSRISK_{it} + \beta_5 LIQUID_{it} + \beta_6 SIZE_{it} + \beta_7 SECTORdummies_{it}$$

Model 3

$$PERFORM_{it} = \alpha_i + \beta_1 INSTOWN_{it} + \beta_2 GROWTH_{it} + \beta_3 LEV_{it} + \beta_4 BUSRISK_{it} + \beta_5 LIQUID_{it} + \beta_6 SIZE_{it} + \beta_7 SECTORdummies_{it}$$

Model 4

$$PERFORM_{it} = \alpha_i + \beta_1 FAMOWN_{it} + \beta_2 GOVOWN_{it} + \beta_3 INSTOWN_{it} + \beta_4 GROWTH_{it}\ \beta_5 LEV_{it} + \beta_6 BUSRISK_{it} + \beta_7 LIQUID_{it} + \beta_8 SIZE_{it} + \beta_9 SECTORdummies$$

The regression results for the hypothesis testing in Table 4 show that family ownership significantly and positively affects company performance. This finding indicates that family ownership improves company performance. These results support those of Ding et al. (2021), Amore et al. (2022), Kraus et al. (2020), Jabbouri and Jabbouri (2020), D'Aurizio et al. (2015), Adams et al. (2008), and Sraer and Thesmar (2007). Amore et al. (2022) compare family companies with non-family companies in Italy. According to the studies, family companies are better positioned to manage employee relations, resulting in leading to increased labor productivity during the COVID-19 pandemic. Companies have a long-term perspective, workforce engagement, and align their relationship with stakeholders. As a result, during the pandemic, they have higher operating profitability and labor productivity. Direct supervision and control by family members strengthen the benefits for their businesses. Kraus et al. (2020) examine family companies in five European countries facing COVID-19. The study results show that family companies from all industries in Indonesia quickly adapted their business models to changing conditions. Additionally, the pandemic introduced significant cultural changes, even though these were not deliberate actions by the company.

Table 4 shows that government ownership negatively affects company performance. This result indicates that government ownership reduces the company's performance. These results support those of Laporšek et al. (2021), Lazzarini and Musacchio (2018), and Tian and Estrin (2008). Although Yu (2013) and Li et al. (2008) find that the company's performance improves with higher government ownership because they have more state support, however, the government's active role weakens corporate governance due to differences in political interests within the company (Laporšek et al. 2021). In China, Tian and Estrin (2008) prove that government ownership can execute more control over the company and divert more assets for political goals because of its concentrated ownership. However, the probability of political interference diminishes once government shareholding reaches a certain level. When the government's ownership style is passive, such as lack of participation in shareholder meetings and inadequate oversight, it weakens management's incentives to maximize state companies' value (Lazzarini and Musacchio 2018). The government has little incentive to assist companies with limited cash flow rights owing to the economic uncertainty caused by the COVID-19 pandemic that affects companies' operational flexibility. Consequently, the preferential treatment or incentives provided by the Government of Indonesia for companies experiencing difficulties may not be realized, thereby reducing their performance.

Table 4 shows that institutional ownership positively affects company performance. This supports previous studies, which show the positive relationship between institutional ownership and company performance (Jabbouri and Jabbouri 2020), (Amin and Hamdan 2018), (Baker et al. 2017), (Abu-Serdaneh et al. 2010), (Shleifer and Vishny 1997). The significant institutional investment in international financial markets makes

an influential player in the global investment business. The high percentage of owner-
ship provides the investors with more voting and control rights to influence manage-
ment to achieve the expected returns. In terms of monitoring outcomes, large insti-
tutional ownership has proven to be an adequate governance mechanism that can re-
duce agency problems and improve company performance during a Moroccan economic
downturn (Jabbouri and Jabbouri 2020) as well as institutional investors overcome the
problem of controlling managers (Amin and Hamdan 2018). Furthermore, large invest-
ments encourage them to abandon passive surveillance and adopt a more active approach
(Shleifer and Vishny 1997). The potential benefits to institutional shareholders of active
management oversight outweigh the associated costs and prevent the potential for signifi-
cant losses (Baker et al. 2017). The study results show that during the COVID-19 pandemic,
large-scale investments improved company performance.

The results of the control variable effect indicate that growth opportunities, leverages,
and firm size positively affect company performance. Because of their long-term cash
flow expectations, companies with greater asset growth are more resilient to the COVID
pandemic (Kaczmarek et al. 2021). Therefore, growth opportunities can improve company
performance. Financial leverage positively affects company performance because it makes
managers more efficient, often referred to as the control hypothesis (Jensen 1986). Managers
make future payments and provide creditors with the right to declare bankruptcy when
they are unable to pay debts. This requires managers to exercise caution in their investments
and maintain their ability to pay principal interests.

There is a positive relationship between firm size and performance. Large companies
are more competitive because of their larger market share and easier capital. Although
the fixed costs of maintaining large companies are high, small companies' compliance
procedures, such as R&D and adoption of new technology are more expensive, complex,
or lengthy for them. In fact, small businesses' viability and growth, frequently face se-
vere financing challenges when compared to large companies. The problems that arise in
financing cause a greater variation in profitability. According to Hendra et al. (2018) in
Indonesia, if the total assets are large, the firm's risk is low since large firms frequently
have multiple sources of funds for supporting their operational activities, both internal
and external. Firms can also use funding sources to generate a profit from their activities.
With high profits, the firm is expected to offer investors high returns. As a result, larger
firms are preferred by investors seeking high returns (Handayani et al. 2019). In addition,
According to Engidaw (2022), small companies are the most affected by the COVID-19 pan-
demic. Levy (2020) found that lockdowns increased the revenue of tech and pharmaceutical
companies. In contrast, the lockdowns hurt and bankrupted smaller companies, which
depended more on the traditional economy. Other findings have shown that the industrial
sector negatively affects company performance. Furthermore, the COVID-19 pandemic
has impacted certain economic sectors more than others have. Shen et al. (2020) find that
tourism and catering are the most affected industrial sectors in China. Bartik et al. (2020)
also find that retail businesses are particularly vulnerable to coronavirus-related disruptions.

Company performance is not significantly affected by liquidity or business risks.
Liquidity is calculated using the current ratio, although a high ratio does not guarantee that
a company can pay its short-term obligations. This is because the inventory held is higher
than the company's ability to increase its sales. Consequently, inventory turnover was
relatively low, consistent with Bartik et al. (2020), who found that the COVID-19 pandemic
had a significant impact on retail companies. Their sales continued to decline, resulting
in a relatively low inventory turnover. Therefore, high liquidity does not guarantee that a
company meets its maturing debts. The impact of liquidity on business performance varies
according to the kind of liquidity calculation and the company. (Suprihati et al. 2018) for
example, indicated that a high liquidity ratio does not affect a company's performance as
long as the cash is derived from significant bad debts. This means there are just asset claims
without physical assets that may be utilized during the epidemic. This might explain why
our data indicate no relationship between liquidity levels and financial success during the

COVID-19 pandemic. The company's management may also explain that this conclusion has less than optimum working capital; as a result, there are still underutilized assets, which may harm profitability since they are burdened. The quantity of liquidity has no bearing on it.

Business risk has no significant effect on company performance. Shareholders are more concerned about corporate governance during economic downturns and resources are more vulnerable to insider takeovers (Mitton 2002). Given the relatively large institutional ownership, the exit strategy of selling shares depresses stock prices and causes substantial losses for investors. Therefore, institutional investors serve as supervisors and managers, so supporting Hypothesis 3. They also reduce risk and improve performance and are willing to pay a premium for firms with significant institutional ownership.

*4.4. Robustness Check*

This section examines whether the previous results are robust to changes in the proxies for independent variables. The check was performed using a dummy variable instead of ownership percentage. According to Anderson and Reeb (2003), the ownership percentage provides a more convincing and controllable measure, but it may reduce or overstate the influence of the company owner. Therefore, a dummy variable was created by dividing the three points of intersection by <20%, 20–50%, and >50%. The aim was to overcome the uncertainty regarding the percentage of ownership size because these three points affect shareholders' control over the company. The following equation was used:

$$\text{PERFORM}_{it} = \alpha_i + \beta_1 \text{FAMOWNdummies}_{it} + \beta_2 \text{GOVOWNdummies}_{it} + \beta_3 \text{INSTOWNdummies}_{it} + \beta_4 \text{GROWTH}_{it}\ \beta_5 \text{LEV}_{it} + \beta_6 \text{BUSRISK}_{it} + \beta_7 \text{LIQUID}_{it} + \beta_8 \text{SIZE}_{it} + \beta_9 \text{SECTORdummies}_{it}$$

Table 6 shows the robustness check results are consistent with prior findings, indicating the effect of the family and institutional ownership coefficients on firm performance is positive and significant. Government ownership has a negative and significant effect on company performance. This finding implies that the previous results are strong for changes in the proxies.

**Table 6.** Robustness check.

|  | Coeff. | Std. Error | t-Value | Prob. |
|---|---|---|---|---|
| C | −0.3444 | 0.2366 | −1.4558 | 0.1458 |
| FAMOWN | 0.0084 | 0.0050 | 1.6776 | 0.0937 * |
| GOVOWN | −0.0631 | 0.0326 | −1.9354 | 0.0532 ** |
| INSTOWN | 0.0170 | 0.0066 | 2.5623 | 0.0105 *** |
| GROWTH | $-3.83 \times 10^{-5}$ | 0.0001 | 0.3564 | 0.7216 |
| LEV | $9.08 \times 10^{-5}$ | 0.0001 | 0.7678 | 0.4428 |
| RISKBUS | −0.0029 | 0.0078 | −0.3678 | 0.7131 |
| LIQUID | $2.70 \times 10^{-5}$ | $9.08 \times 10^{-5}$ | 0.2977 | 0.7660 |
| SIZE | $1.68 \times 10^{-5}$ | $2.34 \times 10^{-6}$ | 7.2078 | 0.0000 |
| SECTOR | −0.0046 | 0.0016 | −2.8519 | 0.0044 |
| Adj R$^2$ | | 0.068 | | |
| F-statistic | | 9.2405 | | |
| Prob F statistic | | 0.0000 | | |
| Observation | | 1025 | | |

Note: ***, **, and * significant at 1%, 5%, and 10%; PERFORM = return on assets ratio, FAMOWN = dummy by dividing the three points of intersection the percentage of family ownership, GOVOWN = dummy by dividing the three points of intersection the percentage of government ownership; INSTOWN = dummy by dividing the three points of intersection the percentage of institutional ownership; LEV = debt to total assets ratio; BURISK = EBIT standard deviation; LIQUID = current ratio; SIZE = Ln total assets; GROWTH = asset growth ratio; SECTOR = dummy for nine industrial sectors in Indonesia. Source: Author's computational results.

## 5. Conclusion, Limitation, and Implications

### 5.1. Conclusion

The COVID-19 pandemic poses threats and opportunities for businesses in various countries including Indonesia. One threat relates to maintaining or improving a company's performance during a pandemic emergency and its relationship to its shareholder identity. As a result, this study aimed to investigate the role of shareholder identity in improving firm performance during shock events, such as the COVID-19 pandemic. The sample comprised non-financial companies that were listed on the Indonesia Stock Exchange (IDX) during 2020–2021. The results show the varied effects of shareholder identity on company performance. Family and institutional ownership significantly and positively affected company performance, whereas government ownership had a negative impact.

The findings revealed that family and institutional ownership positively affected company performance during the COVID-19 pandemic. This is because family businesses have better operating profitability and productivity. Furthermore, direct supervision and control by family members strengthen the benefits provided by families to their businesses throughout the pandemic. High institutional ownership makes investors play a substantial role in reducing business risk and improving company performance. This strengthens the findings on the positive significance of institutional ownership in company performance. Furthermore, government ownership negatively affects a company's performance. As a majority shareholder, the government has different strategic goals oriented toward better public services than financial gains.

It is important to consider how shareholder involvement affects a company's performance, especially during the pandemic, because the treatment is different. This study contributes to the evidence regarding the COVID-19 pandemic. It is expected to help regulatory agencies, policymakers, and companies develop public and corporate governance policies for future emergency preparedness and response.

### 5.2. Limitation

Finally, we acknowledge the limitations of our work, which provide opportunities for further research. Because institutional investors are not homogeneous, necessitating future studies to differentiate between the types of institutional investors. The shareholders are classified as active, or passive based on their monitoring power. Along with differences in owner types, their monitoring responsibilities are likely to differ. These investors have different strategies and interests that pressure the company and its stakeholders. Individual owners are always less active in controlling activities than institutional owners. This is due to the difficulty of individuals intervening collectively due to the high cost of ownership. However, because of the trade-off between the benefits and the costs of active oversight, institutional investors may not have the same incentive to improve corporate governance for the two reasons listed below. To begin with, institutional investors' portfolios are heterogeneous. Differences in investment horizons, objectives, activity levels, and sizes explain the varying participation and activity levels of institutional investors (Larcker and Tayan 2011). Second, investors' attention is limited. They are unable to monitor all of the companies in their portfolio. Therefore, as a result, institutional oversight is motivated by the importance of specific stock in their portfolio. Therefore, future research may categorize institutional investors by size or type to examine their separate impacts on business performance during a crisis. Cross-country comparisons may also be used to identify and evaluate the influence of macroeconomic and country-specific factors on business success. The present research can be re-examined using another set of CG mechanisms (e.g., managerial ownership, dispersed ownership, ownership concentration, etc.) and the results compared with the current developments; Despite these limitations, this study provides new evidence regarding the COVID-19 global outbreak. This can assist managers, regulators, and other stakeholders in formulating future corporate governance strategies regarding emergency preparedness and response.

### 5.3. Implication

These findings provide important theoretical and practical implications that are useful to academic researchers, company managers, investors, and regulators.

According to our findings, organizations with considerable family and institutional ownership face a high level of resistance to sustaining their business firm performance. This finding is consistent with that of (Perrini et al. 2008) and (Al-Janadi 2021) both of which found that internal ownership is positively related to organizational performance. Both are doing research in non-centralized companies. Furthermore, Al-Janadi (2021) found that non-financial firms moderated their central ownership relationship with firm performance, moving from a negative to a significantly positive relationship. This weakness stems from the weakening role of internal ownership as non-financial firms have a higher concentration of ownership and more control. Our research provides additional niche value and proves that ownership structure influences company performance, suggesting that future research will test more niches for it. In practice, our research suggests that enterprises should retain the ratio of family ownership to maintain direct supervision of the family in order to boost productivity throughout the pandemic. Based on the findings of this study, organizations should consider decreasing company risk through institutional ownership. Another practical relevance of the findings for the control variables is the requirement for enterprises to monitor the graph of asset growth and financial leverage on a frequent basis and to carry out appropriate actions.

Regulators can evaluate governance issues, review procedures, and regulations, and take necessary steps to strengthen investor protection and restore capital market integrity. Strong investor protection and improved corporate governance boost investor trust, stimulate market activity, and contribute to economic growth. Furthermore, in order to increase their understanding of capital markets, regulators must comprehend the influence of their investment decisions on the financial performance of issuers to improve their understanding of capital markets. This study suggests that companies act responsively and adaptively to the business environment uncertainties faced through several mechanisms such as good governance. In addition, the government needs to establish appropriate monetary and economic policies to mitigate the adverse consequences of the next crisis.

**Author Contributions:** Conceptualization, D.P.; methodology, D.P. and A.N.; software, D.P.; validation, D.S., A.N. and I.P.R.; formal analysis, D.P. and D.S.; investigation, D.P. and I.P.R.; writing—original draft preparation, D.P. and A.N., writing—review and editing, D.S. and I.P.R.; supervision, D.P. All authors have read and agreed to the published version of the manuscript.

**Funding:** This research was funded by the Research University Grant of Sebelas Maret University, Number 254/UN27.22/PT.01.03/2022, and The APC was funded by the Research University Grant of Sebelas Maret University.

**Institutional Review Board Statement:** Not applicable.

**Informed Consent Statement:** Not applicable.

**Data Availability Statement:** Not applicable.

**Conflicts of Interest:** The authors declare no conflict of interest.

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
