# Peer review of "Firm Performance during COVID-19 Pandemic: Does Ownership Identity Matter? Evidence from Indonesia"

_jrfm, doi:10.3390/jrfm15100444_

Round 1

Reviewer 1 Report

I can present the following comments:

1- In the abstract, you should add recommendations at the end of the abstract.

2- In the results, your must clearly identify the result of each hypothesis and which of the previous studies results support each of your hypothesis clearly. For example, in the results, i did not see any comments aboiut the results or supporting results by other studies regarding the second hypothesis of government ownership.

3- Clearly mention which previous studies support this or that of your hypothesis' result.

 4- The study sample consists of all companies listed in Indonesia Bourse. Why you did not depand on only sample of the coumunity? The usual procesure is to use only sample of the comunity.

All the best,

Reviewer 2 Report

    Dear Authors, I am sending comments to the article: „Firm Performance During Covid-19 Pandemic:
Does Ownership Identity Matter?”

1.

„Return on Assets (ROA) is commonly used in accounting-based measurements (Al-Matari et al. 2014). This metric reflects a company’s efficiency in using assets in operational activities to meet shareholders’ economic interests.”

Why did the authors use this ROA in their research?
Profitability of sales seems to me to be an important indicator at the time of Covid?

2.

Line 420 „There is a positive relationship between firm size and performance.
Large
companies  are more competitive because of their larger market share and better access to capital.” In general, large companies are characterized by lower profitability compared
to small companies. The fixed costs of maintaining such companies are high.
It also depends on the industry.
Was there such a relationship in the surveyed companies?

3.

Line 433-435 „Company performance is not significantly affected by
liquidity or
business risks. Liquidity is calculated using the current ratio,
although a
high ratio does not guarantee that  a company can pay
its short-term obligations. This is because the inventory held
is higher than the company's ability to increase its sales”

Line 436-439 „Consequently, inventory turnover was relatively low, consistent with Bartik et al. (2020), who found that the COVID-19 pandemic severely impacted retail companies. Their sales continued to decline, resulting in a relatively low inventory turnover. Therefore, high liquidity does not guarantee that a company meets its maturing debts.

Very interesting conclusions.
During Covid, many companies built up high stocks because managers were concerned about supply disruptions.
Supply problems have occurred in many sectors in Europe.
What later resulted in a quick stock rotation and high profitability.
Below are articles describing such a situation:
Zimon, Grzegorz, Vitalina Babenko, Beata Sadowska, Katarzyna Chudy-Laskowska, and Blanka Gosik. 2021.
"Inventory Management in SMEs Operating in Polish Group Purchasing Organizations during the COVID-19 Pandemic"
Risks 9, no. 4: 63. https://doi.org/10.3390/risks9040063
Zimon, Grzegorz, and Hossein Tarighi. 2021.
"Effects of the COVID-19 Global Crisis on the Working Capital Management Policy: Evidence from Poland"
Journal of Risk and Financial Management 14, no. 4: 169. https://doi.org/10.3390/jrfm14040169
I understand that in the case of the analyzed companies there was no such situation?
What was it caused by? Please refer to this situation in the article.
During COVID, companies that operated continuously had high liquidity.
In the case of the analyzed companies in the article ,
the level of liquidity was influenced by short-term receivables and short-term liabilities ?
I request information.

Sincerely

Reviewer 3 Report

The authors should consider the following recommendations in order to improve the original manuscript:

- This research study is a case study for Indonesia and should it be mentioned in the title as not to generate confusion.

- The keywords should not overlap with the title of this paper.

- To include certain relevant research questions.

- To include the structure of the paper in the Introduction section.

- Authors should take into consideration much more recent publications in the sphere of discussed subject matter, especially studies conducted during the last 2 years. Please discuss more about Covid-19 pandemic caused by Severe Acute Respiratory Syndrome Coronavirus 2 (SARS-CoV-2) and its impact on economy, financial and social environment. I suggest extending the literature section by including recent and relevant studies, such as for instance:

a) P. Wanke, M. A. K. Azad, A. Karbassi Yazdi, F. R. Birau and C. M. Spulbar, "Revisiting CAMELS Rating System and the Performance of ASEAN Banks: A Comprehensive MCDM/Z-Numbers Approach," in IEEE Access, vol. 10, pp. 54098-54109, 2022, doi: 10.1109/ACCESS.2022.3171339.

b) Pourmansouri R, Mehdiabadi A, Shahabi V, Spulbar C, Birau R. An Investigation of the Link between Major Shareholders’ Behavior and Corporate Governance Performance before and after the COVID-19 Pandemic: A Case Study of the Companies Listed on the Iranian Stock Market. Journal of Risk and Financial Management. 2022; 15(5):208. https://doi.org/10.3390/jrfm15050208.

- Deepen the description of the limitations of conducted research and indicate the trends for further empirical research.

-         To expand the managerial implications in the article.

-        The sources must be added under each table.

-           Human proofreading, English grammar and spelling correction are also required in order to improve the quality of the manuscript.

-          I would also like to see a well-developed discussion comparing and contrasting solution/results presented in the work with existing work and then a subsection of it presenting contributions to theory/knowledge/literature and followed by a subsection on “Implications for practice”.

Round 2

Reviewer 3 Report

The cited references must also be included in the References section (see for instance Pourmansori et al. ,2022).

Author Response

To the honorable reviewer 3
We would like to give a response for your review below:

"The cited references must also be included in the References section (see for instance Pourmansori et al. ,2022)."

We have already added the respective reference to our manuscript attached below.

Best Regards
